# Long-Term Weight-Loss Lifestyle Modification Programme in a Patient with Severe Lumbar Intervertebral Disc Degeneration and Obesity: A Case Report

**Dana El Masri [1], Dima Kreidieh [1], Hana Tannir [1], Leila Itani [1] and Marwan El Ghoch [1,2,\*]**

[1]  Department of Nutrition and Dietetics, Faculty of Health Sciences, Beirut Arab University, P.O. Box 11-5020 Riad El Solh, Beirut 11072809, Lebanon; dana.masri@bau.edu.lb (D.E.M.); d.kraydeyeh@bau.edu.lb (D.K.); hana.tannir@bau.edu.lb (H.T.); l.itani@bau.edu.lb (L.I.)

[2]  Nutrition and Dietetics Program—CTR Centre, 42124 Reggio Emilia, Italy

\*  Correspondence: m.elghoch@bau.edu.lb or marwan1979@hotmail.com; Tel.: +39-0522-385-411; Fax: +39-0522-385-473

**Abstract:** Obesity is a growing health problem worldwide, associated with serious medical and psychosocial comorbidities that increase the risk of mortality. Strong evidence confirms lifestyle modification programmes as the cornerstone of its treatment. However, the available long-term lifestyle modification programmes for weight management delivered in Arabic-speaking countries seem to be lacking in effectiveness in terms of weight-loss maintenance and do not conform to the standard for clinical significance. Factors such as methodological weaknesses in programme transcultural adaptation and the lack of expert clinical supervision before and during implementation seem to underlie this discrepancy. In this case report, we describe for the first time an Arabic-speaking patient with obesity and severe lumbar intervertebral disc degeneration, who successfully underwent weight management by means of a new, well-adapted and well-implemented personalized cognitive behavioural programme for obesity (CBT-OB). After eighteen months, the patient displayed significant weight-loss maintenance (~16% weight-loss), improvement in total and central body fat distribution, reduced pain from disc degeneration, and an increase in high-density lipoprotein (HDL). The CBT-OB programme may be a feasible approach to managing Arab patients with obesity, producing long-lasting weight-loss maintenance improvements in the obesity-related profile.

**Keywords:** obesity; CBT-OB; lifestyle modification; cognitive behavioural therapy; weight loss treatment; Arab countries

## 1. Introduction

Obesity is a significant health condition associated with non-communicable diseases [1,2], psychosocial comorbidities [3], poor health-related quality of life [4] and an increased rate of mortality [5]. Moreover, an increasing prevalence of obesity worldwide, including in Arabic-speaking countries, has been reported [6]. Hence, international guidelines strongly recommend evidence-based prevention and treatment approaches to stop the rise in obesity and its associated comorbidities [7].

Strong evidence confirms that long-term lifestyle modification programmes based on behavioural or cognitive behavioural treatment combined with specific recommendations on diet and exercise, are considered to be the cornerstones of the treatment of obesity [7]. They show improvements in weight-related comorbidities and are therefore recommended as first-choice treatments by international guidelines [7].

A recent systematic review and meta-analysis assessed the effectiveness of long-term lifestyle modification programmes for weight management delivered in Arabic-speaking countries, and found that these programmes were not effective in terms of weight-loss maintenance (= 4%). Also, they did not conform to the standard for clinical significance (> 10%) [8]. Authors showed evidence that this may be due to methodological weaknesses in the transcultural adaptation of these programmes, and the lack of expert clinical supervision before and during implementation [8].

In this case report, we describe for the first time an Arabic-speaking patient with obesity and severe lumbar intervertebral disc degeneration, who successfully underwent weight management by means of a well-adapted and well-implemented personalized cognitive behavioural programme for obesity (CBT-OB). After one and half years, the patient displayed significant weight-loss maintenance and significant improvement in total body fat and distribution, plus an increase in high-density lipoprotein (HDL), known to be associated with a lower risk of heart disease.

## 2. Case Presentation

With the consent of the patient, we present the case of a 49-year-old female with class I obesity (body weight = 86.9 kg; height = 162.5 cm; BMI = 32.9 kg/m$^2$) who came to the BAU Clinic for Nutritional and Weight Management at the Beirut Arab University (Lebanon) in February 2017. On admission, a questionnaire was administered in order to retrieve information regarding her medical history, lifestyle (eating habits, levels of physical activity and smoking) and demographic and social conditions (age, gender and marital status).

Accordingly, the patient was diagnosed with severe lumbar intervertebral disc degeneration with diffuse bulge, as identified by magnetic resonance imaging (MRI) in March 2017, using a multiple sagittal and axial T1 and T2 sequences. The examination was performed on a 3 Tesla, which revealed a large central L2–L3 disc herniation with upward migration compromising the spinal canal, dural sac, and neural elements that explained the painful symptomatology (Figure 1A), with urgent indications for surgical intervention. No other diseases (i.e., coronary heart disease, stroke, metabolic diseases) were reported.

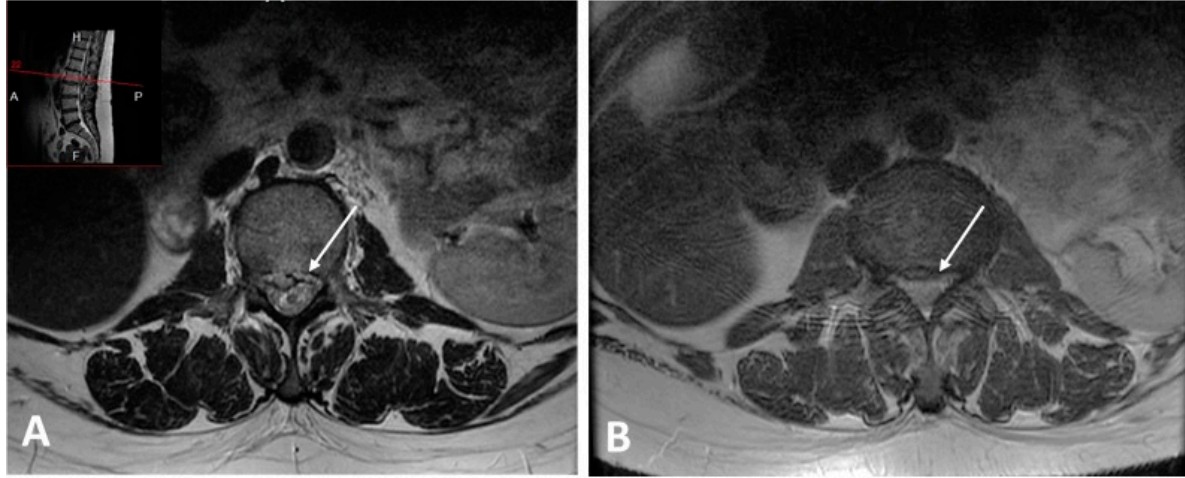

**Figure 1.** MRI image before weight loss shows a large central L2–L3 disc herniation compromising the spinal canal (**A**). After weight loss a significant regression of the large L2–L3 disc extrusion (**B**).

With regard to body weight history, over the past 20 years before coming to our attention, the patient's body weight had progressively increased by about 15 kg to just over 86 kg (body weight at age 20 = 71 kg), which indicated the onset of obesity. This appeared to be the result of a sedentary lifestyle enforced by severe lumbar intervertebral disc degeneration. She reported having made several weight-loss attempts (>10 times), with unsuccessful results. Finally, she reported no

binge-eating episodes or purging over the preceding four weeks, which indicated the absence of a binge-eating disorder.

On admission, her body weight was measured to the nearest 0.1 kg using electronic weighing scales (SECA 2730-ASTRA). Height was measured to the nearest 0.5 cm using a stadiometer. The BMI of the participant was then determined according to the standard formula of body weight (kg) divided by height (m) squared (weight = 86.9 kg; BMI = 32.9 kg/m$^2$). The body composition patterns and distribution were measured using a body composition analyser (InBody 170), which provides separate body-mass readings for different segments of the body, and uses an algorithm incorporating impedance, age and height to estimate skeletal muscle mass (SMM = 25.6 kg), total fat mass percentage (FM% = 46.1%), trunk fat percentage (46.1%) and visceral fat (level = 14). The patient's fasting blood samples and laboratory tests are shown in Table 1.

**Table 1.** Body composition, resting energy expenditure variables, and laboratory results on the morning after overnight fasting.

| Variable | 1st Assessment Before Treatment | 2nd Assessment at the End of Treatment (18 Month Follow-Up) |
|---|---|---|
| Body weight (kg) | 86.9 | 73.5 |
| Skeletal muscle mass (kg) | 25.6 | 24.5 |
| BMI (kg/m$^2$) | 32.9 | 27.8 |
| Total Fat mass percentage (%) | 46.1 | 38.1 |
| Trunk fat percentage (%) | 46.1 | 39.6 |
| Visceral fat level | 14 | 9 |
| Estimated Basal Metabolic Rate (kcal/d) | 1382 | 1353 |
| Total cholesterol (mg/dL) | 163 | 161 |
| HDL cholesterol (mg/dL) | 55 | 65 |
| LDL cholesterol (mg/dL) | 94 | 86.6 |
| Triglycerides (mg/dL) | 67 | 47 |
| Fasting Glucose (mg/dL) | 85 | 85 |
| HbA1c (%) | 4.80 | 4.88 |

## 3. Personalized Multi-Step Cognitive Behavioural Therapy for Obesity (CBT-OB)

This programme featured a low-calorie diet of 1100 kcal/day. The protocol of the treatment essentially involves a personalized CBT-OB programme designed for patients with obesity, which is described in detail in its dedicated English-language treatment manual [9]. In brief, CBT-OB combines specific physical activity and dietary strategies with cognitive behavioural procedures [9]. The programme includes two phases. Phase 1 (weight-loss) has a duration of six months and Phase 2 (weight maintenance) includes monthly visits for a period of 12 months. Each session lasts for 30 min, during which the patient receives the six modules of the programme: (1) monitoring food intake, (2) changing eating habits, (3) developing an active lifestyle, (4) addressing obstacles to weight loss, (5) addressing weight loss and primary goals, and (6) addressing obstacles to weight maintenance [9].

## 4. Treatment Outcomes

The patient completed the CBT-OB programme on 12 September 2018. At this time, her body weight had fallen to 73.5 kg and her BMI to 27.8 kg/m$^2$—a weight loss of 16%, and there was a clear improvement in pain symptoms from intervertebral disc degeneration. This was corroborated by MRI imaging repeated in March 2018, which revealed a significant regression of the large L2–L3 disc extrusion (Figure 1B). On that basis, the consultant surgeon gave no further indications for intervention. The InBody body composition assessment was repeated and revealed improvements, with a significant decrease in total and central body fat (FM% = 38.1%, trunk fat percentage = 39.6%, visceral fat level = 9) and no deterioration in skeletal mass (SMM = 24.5 kg) (Table 1). The blood test revealed a significant increase in high-density lipoprotein (HDL) cholesterol (65 mg/dL; normal values of >55 mg/dL), which is known to be associated with a lower risk of heart disease [10] (Table 1).

## 5. Discussion

To the best of our knowledge, this is the first case of weight management in an Arabic-speaking adult patient with obesity by means of a well-adapted and implemented, long-term personalized lifestyle modification programme based on cognitive behavioural therapy for obesity (CBT-OB) in an Arab country. The importance of our case report stems from the fact that the other long-term lifestyle modification programmes for weight management, delivered in Arabic-speaking countries, seem to ineffective [8]. This programme includes several procedures that highlight the importance of changing eating habits, physical activity and cognitive change [9]. Its main purpose is to help patients achieve and maintain a healthy weight loss [9]. In our patient, nearly 16% weight loss via the CBT-OB lifestyle modification programme resulted in significant weight-loss maintenance over a reasonable period (18 months), reduction of total and central fat, improvement in HDL-C, known to reduce risk of heart disease and improvement in pain symptomatology of intervertebral disc degeneration (back pain and leg pain) which prevented surgical intervention. The results achieved by our patient are in line with those derived from other patients that underwent the same lifestyle modification programme that used the original version of the treatment [11].

Since it is uncertain whether evidence-based treatments developed within a particular context are applicable to different populations with different languages, cultures and values, we took into account linguistic and socio-economic factors in order to avoid the substantial methodological weaknesses usually seen during transcultural adaptation and development of behavioural treatments, e.g., lifestyle modification programmes [12]. Moreover, our group received on-site, weekly direct supervision via Skype and Email by a clinical expert (M.E.) in CBT-OB lifestyle modification programmes, either before or during programme implementation, which ensured the internal validity of the treatment [13].

That being said, the data gathered relates to only one patient, and further data derived from a greater number of patients will be necessary to confirm the effectiveness and efficacy of CBT-OB in Arabic-speaking countries. Nevertheless, our results do indicate that a personalized CBT-OB programme, administered over a period of a year and a half, may be a feasible approach to managing Arab patients with obesity, producing significant weight loss, weight-loss maintenance and improvements in the obesity-related profile (clinical condition and body composition patterns).

**Author Contributions:** All authors claim authorship, and have approved and made substantial contributions to the conception, drafting, and final version of the paper. The study was designed by M.E.G. The data were collected by D.E.M., D.K., and H.T. The manuscript was co-wrote by M.E.G., D.E.M., D.K., H.T. and L.I.

**Funding:** This research received no external funding.

**Acknowledgments:** The authors are grateful to the patient for her cooperation in preparing this case report.

**Conflicts of Interest:** The authors declare no conflict of interest.

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
