# Peer review of "Long-Term Weight-Loss Lifestyle Modification Programme in a Patient with Severe Lumbar Intervertebral Disc Degeneration and Obesity: A Case Report"

_reports, doi:10.3390/reports1030021_

Round 1

Reviewer 1 Report

I think this is a very interesting and well-written case report that will add to the obesity literature.

I just have a couple of very minor suggestions:

In the abstract:

line 29, would change "16%" to "16% weight-loss"

line 30, would change "pain symptoms from intervertebral disc degeneration"  to "reduced pain from disc degeneration"

In the discussion

I think it might be nice in the discussion to mention any particular challenges that arose in adapting the program to an Arabic speaking patient.

Author Response

In the abstract:

Line 29, would change "16%" to "16% weight-loss"

Response: Done as suggested  (Line 31).

Line 30, would change "pain symptoms from intervertebral disc degeneration" 

to "reduced pain from disc degeneration"

Response: Done as suggested  (Line 32).

In the discussion

I think it might be nice in the discussion to mention any particular challenges that arose in adapting the program to an Arabic speaking patient.

Response: Done as suggested (Lines 124-126).

Reviewer 2 Report

The authors describe a case of an Arabic-speaking patient with obesity and severe lumbar intervertebral disc degeneration, who successfully underwent weight management by means of a new, well-adapted and well-implemented personalized cognitive behavioural programme for obesity (CBT-OB). The study is not enough innovative but it presents the first data from Arabic country where studies maybe rare. This corresponds more to a short note than a scientific article. However, there are some problem had to be detail clarified

1. The consent of patient must be reported.

2. Other metabolic parameters, such as HbA1C, is nessary to follow up the obesity

3. Please, provide the past history or medicine history, including coronary heart disease, stroke, metabolic diseaes. 

4. Please, provide a flowchart of CBT-OB for readers.

5. Extensive editing of English language s necessary.

Author Response

The authors describe a case of an Arabic-speaking patient with obesity and severe lumbar intervertebral disc degeneration, who successfully underwent weight management by means of a new, well-adapted and well-implemented personalized cognitive behavioural programme for obesity (CBT-OB). The study is not enough innovative but it presents the first data from Arabic country where studies maybe rare. This corresponds more to a short note than a scientific article. However, there are some problem had to be detail clarified

1. The consent of patient must be reported.

Response: Done as suggested. The statement “With the consent of the patient” has been added (Line 65).

2. Other metabolic parameters, such as HbA1C, is necessary to follow up the obesity

Response: Done as suggested. We added other metabolic parameters (Triglycerides and HbA1c) in table 1.

3. Please, provide the past history or medicine history, including coronary heart disease, stroke, metabolic diseases. 

Response: Done as suggested (Line 76-77). 

4. Please, provide a flowchart of CBT-OB for readers.

Response: The aim of our paper is not to describe the treatment that has been extensively explained in the treatment developer book and many other published papers.

5. Extensive editing of English language s necessary.

Response: The manuscript has been corrected initially by English professional editor, however we gone for a second round editing of the revised version.